# (More) Efficient Reinforcement Learning via Posterior Sampling

**Osband, Ian**
Stanford University
Stanford, CA 94305
iosband@stanford.edu

**Van Roy, Benjamin**
Stanford University
Stanford, CA 94305
bvr@stanford.edu

**Russo, Daniel**
Stanford University
Stanford, CA 94305
djrusso@stanford.edu

## Abstract

Most provably-efficient reinforcement learning algorithms introduce optimism about poorly-understood states and actions to encourage exploration. We study an alternative approach for efficient exploration: *posterior sampling for reinforcement learning* (PSRL). This algorithm proceeds in repeated episodes of known duration. At the start of each episode, PSRL updates a prior distribution over Markov decision processes and takes one sample from this posterior. PSRL then follows the policy that is optimal for this *sample* during the episode. The algorithm is conceptually simple, computationally efficient and allows an agent to encode prior knowledge in a natural way. We establish an $\tilde{O}(\tau S \sqrt{AT})$ bound on expected regret, where $T$ is time, $\tau$ is the episode length and $S$ and $A$ are the cardinalities of the state and action spaces. This bound is one of the first for an algorithm not based on optimism, and close to the state of the art for any reinforcement learning algorithm. We show through simulation that PSRL significantly outperforms existing algorithms with similar regret bounds.

## 1 Introduction

We consider the classical reinforcement learning problem of an agent interacting with its environment while trying to maximize total reward accumulated over time [1, 2]. The agent's environment is modeled as a Markov decision process (MDP), but the agent is uncertain about the true dynamics of the MDP. As the agent interacts with its environment, it observes the outcomes that result from previous states and actions, and learns about the system dynamics. This leads to a fundamental tradeoff: by exploring poorly-understood states and actions the agent can learn to improve future performance, but it may attain better short-run performance by exploiting its existing knowledge.

Naïve optimization using point estimates for unknown variables overstates an agent's knowledge, and can lead to premature and suboptimal exploitation. To offset this, the majority of provably efficient learning algorithms use a principle known as *optimism in the face of uncertainty* [3] to encourage exploration. In such an algorithm, each state and action is afforded some optimism bonus such that their value to the agent is modeled to be as high as is statistically plausible. The agent will then choose a policy that is optimal under this "optimistic" model of the environment. This incentivizes exploration since poorly-understood states and actions will receive a higher optimism bonus. As the agent resolves its uncertainty, the effect of optimism is reduced and the agent's behavior approaches optimality. Many authors have provided strong theoretical guarantees for optimistic algorithms [4, 5, 6, 7, 8]. In fact, almost all reinforcement learning algorithms with polynomial bounds on sample complexity employ optimism to guide exploration.

We study an alternative approach to efficient exploration, *posterior sampling*, and provide finite time bounds on regret. We model the agent's initial uncertainty over the environment through a prior distribution.[1] At the start of each *episode*, the agent chooses a new policy, which it follows for the duration of the episode. Posterior sampling for reinforcement learning (PSRL) selects this policy through two simple steps. First, a single instance of the environment is sampled from the posterior distribution at the start of an episode. Then, PSRL solves for and executes the policy that is optimal under the sampled environment over the episode. PSRL randomly selects policies according to the probability they are optimal; exploration is guided by the variance of sampled policies as opposed to optimism.

The idea of posterior sampling goes back to 1933 [9] and has been applied successfully to multi-armed bandits. In that literature, the algorithm is often referred to as *Thompson sampling* or as *probability matching*. Despite its long history, posterior sampling was largely neglected by the multi-armed bandit literature until empirical studies [10, 11] demonstrated that the algorithm could produce state of the art performance. This prompted a surge of interest, and a variety of strong theoretical guarantees are now available [12, 13, 14, 15]. Our results suggest this method has great potential in reinforcement learning as well.

PSRL was originally introduced in the context of reinforcement learning by Strens [16] under the name "Bayesian Dynamic Programming",[2] where it appeared primarily as a heuristic method. In reference to PSRL and other "Bayesian RL" algorithms, Kolter and Ng [17] write "little is known about these algorithms from a theoretical perspective, and it is unclear, what (if any) formal guarantees can be made for such approaches." Those Bayesian algorithms for which performance guarantees exist are guided by optimism. BOSS [18] introduces a more complicated version of PSRL that samples many MDPs, instead of just one, and then combines them into an *optimistic* environment to guide exploration. BEB [17] adds an exploration bonus to states and actions according to how infrequently they have been visited. We show it is not always necessary to introduce optimism via a complicated construction, and that the simple algorithm originally proposed by Strens [16] satisfies strong bounds itself.

Our work is motivated by several advantages of posterior sampling relative to optimistic algorithms. First, since PSRL only requires solving for an optimal policy for a single sampled MDP, it is computationally efficient both relative to many optimistic methods, which require simultaneous optimization across a *family* of plausible environments [4, 5, 18], and to computationally intensive approaches that attempt to approximate the Bayes-optimal solutions directly [18, 19, 20]. Second, the presence of an explicit prior allows an agent to incorporate known environment structure in a natural way. This is crucial for most practical applications, as learning without prior knowledge requires exhaustive experimentation in each possible state. Finally, posterior sampling allows us to separate the *algorithm* from the *analysis*. In any optimistic algorithm, performance is greatly influenced by the manner in which optimism is implemented. Past works have designed algorithms, at least in part, to facilitate theoretical analysis for toy problems. Although our analysis of posterior sampling is closely related to the analysis in [4], this worst-case bound has no impact on the algorithm's actual performance. In addition, PSRL is naturally suited to more complex settings where design of an efficiently optimistic algorithm might not be possible. We demonstrate through a computational study in Section 6 that PSRL outperforms the optimistic algorithm UCRL2 [4]: a competitor with similar regret bounds over some example MDPs.

## 2   Problem formulation

We consider the problem of learning to optimize a random finite horizon MDP $M = (\mathcal{S}, \mathcal{A}, R^M, P^M, \tau, \rho)$ in repeated finite episodes of interaction. $\mathcal{S}$ is the state space, $\mathcal{A}$ is the action space, $R_a^M(s)$ is a probability distribution over reward realized when selecting action $a$ while in state $s$ whose support is $[0, 1]$, $P_a^M(s'|s)$ is the probability of transitioning to state $s'$ if action $a$ is selected while at state $s$, $\tau$ is the time horizon, and $\rho$ the initial state distribution. We define the MDP and all other random variables we will consider with

respect to a probability space $(\Omega, \mathcal{F}, \mathbb{P})$. We assume $\mathcal{S}$, $\mathcal{A}$, and $\tau$ are deterministic so the agent need not learn the state and action spaces or the time horizon.

A deterministic policy $\mu$ is a function mapping each state $s \in \mathcal{S}$ and $i = 1, \ldots, \tau$ to an action $a \in \mathcal{A}$. For each MDP $M = (\mathcal{S}, \mathcal{A}, R^M, P^M, \tau, \rho)$ and policy $\mu$, we define a value function

$$V_{\mu,i}^M(s) := \mathbb{E}_{M,\mu}\left[\sum_{j=i}^{\tau} \overline{R}_{a_j}^M(s_j)\Big| s_i = s\right],$$

where $\overline{R}_a^M(s)$ denotes the expected reward realized when action $a$ is selected while in state $s$, and the subscripts of the expectation operator indicate that $a_j = \mu(s_j, j)$, and $s_{j+1} \sim P_{a_j}^M(\cdot|s_j)$ for $j = i, \ldots, \tau$. A policy $\mu$ is said to be optimal for MDP $M$ if $V_{\mu,i}^M(s) = \max_{\mu'} V_{\mu',i}^M(s)$ for all $s \in \mathcal{S}$ and $i = 1, \ldots, \tau$. We will associate with each MDP $M$ a policy $\mu^M$ that is optimal for $M$.

The reinforcement learning agent interacts with the MDP over episodes that begin at times $t_k = (k-1)\tau + 1$, $k = 1, 2, \ldots$. At each time $t$, the agent selects an action $a_t$, observes a scalar reward $r_t$, and then transitions to $s_{t+1}$. If an agent follows a policy $\mu$ then when in state $s$ at time $t$ during episode $k$, it selects an action $a_t = \mu(s, t - t_k)$. Let $H_t = (s_1, a_1, r_1, \ldots, s_{t-1}, a_{t-1}, r_{t-1})$ denote the history of observations made *prior* to time $t$. A reinforcement learning algorithm is a deterministic sequence $\{\pi_k | k = 1, 2, \ldots\}$ of functions, each mapping $H_{t_k}$ to a probability distribution $\pi_k(H_{t_k})$ over policies. At the start of the $k$th episode, the algorithm samples a policy $\mu_k$ from the distribution $\pi_k(H_{t_k})$. The algorithm then selects actions $a_t = \mu_k(s_t, t - t_k)$ at times $t$ during the $k$th episode.

We define the regret incurred by a reinforcement learning algorithm $\pi$ up to time $T$ to be

$$\text{Regret}(T, \pi) := \sum_{k=1}^{\lceil T/\tau \rceil} \Delta_k,$$

where $\Delta_k$ denotes regret over the $k$th episode, defined with respect to the MDP $M^*$ by

$$\Delta_k = \sum_{s \in \mathcal{S}} \rho(s)(V_{\mu^*,1}^{M^*}(s) - V_{\mu_k,1}^{M^*}(s)),$$

with $\mu^* = \mu^{M^*}$ and $\mu_k \sim \pi_k(H_{t_k})$. Note that regret is not deterministic since it can depend on the random MDP $M^*$, the algorithm's internal random sampling and, through the history $H_{t_k}$, on previous random transitions and random rewards. We will assess and compare algorithm performance in terms of regret and its expectation.

## 3 Posterior sampling for reinforcement learning

The use of posterior sampling for reinforcement learning (PSRL) was first proposed by Strens [16]. PSRL begins with a prior distribution over MDPs with states $\mathcal{S}$, actions $\mathcal{A}$ and horizon $\tau$. At the start of each $k$th episode, PSRL samples an MDP $M_k$ from the posterior distribution conditioned on the history $H_{t_k}$ available at that time. PSRL then computes and follows the policy $\mu_k = \mu^{M_k}$ over episode $k$.

---

**Algorithm: Posterior Sampling for Reinforcement Learning (PSRL)**

---

**Data**: Prior distribution $f$, t=1
**for** *episodes* $k = 1, 2, \ldots$ **do**
    sample $M_k \sim f(\cdot|H_{t_k})$
    compute $\mu_k = \mu^{M_k}$
    **for** *timesteps* $j = 1, \ldots, \tau$ **do**
        sample and apply $a_t = \mu_k(s_t, j)$
        observe $r_t$ and $s_{t+1}$
        $t = t + 1$
    **end**
**end**

---

We show PSRL obeys performance guarantees intimately related to those for learning algorithms based upon OFU, as has been demonstrated for multi-armed bandit problems [15]. We believe that a posterior sampling approach offers some inherent advantages. Optimistic algorithms require explicit construction of the confidence bounds on $V_{\mu,1}^{M^*}(s)$ based on observed data, which is a complicated statistical problem even for simple models. In addition, even if strong confidence bounds for $V_{\mu,1}^{M^*}(s)$ were known, solving for the best optimistic policy may be computationally intractable. Algorithms such as UCRL2 [4] are computationally tractable, but must resort to separately bounding $\overline{R}_a^M(s)$ and $P_a^M(s)$ with high probability for each $s, a$. These bounds allow a "worst-case" mis-estimation simultaneously in *every* state-action pair and consequently give rise to a confidence set which may be far too conservative.

By contrast, PSRL always selects policies according to the probability they are optimal. Uncertainty about each policy is quantified in a statistically efficient way through the posterior distribution. The algorithm only requires a single sample from the posterior, which may be approximated through algorithms such as Metropolis-Hastings if no closed form exists. As such, we believe PSRL will be simpler to implement, computationally cheaper and statistically more efficient than existing optimistic methods.

## 3.1 Main results

The following result establishes regret bounds for PSRL. The bounds have $\tilde{O}(\tau S \sqrt{AT})$ expected regret, and, to our knowledge, provide the first guarantees for an algorithm not based upon optimism:

**Theorem 1.** *If $f$ is the distribution of $M^*$ then,*

$$\mathbb{E}\left[\text{Regret}(T, \pi_\tau^{\text{PS}})\right] = O\left(\tau S \sqrt{AT \log(SAT)}\right) \tag{1}$$

This result holds for *any* prior distribution on MDPs, and so applies to an immense class of models. To accommodate this generality, the result bounds expected regret under the prior distribution (sometimes called *Bayes risk* or *Bayesian regret*). We feel this is a natural measure of performance, but should emphasize that it is more common in the literature to bound regret under a worst-case MDP instance. The next result provides a link between these notions of regret. Applying Markov's inequality to (1) gives convergence in probability.

**Corollary 1.** *If $f$ is the distribution of $M^*$ then for any $\alpha > \frac{1}{2}$,*

$$\frac{\text{Regret}(T, \pi_\tau^{\text{PS}})}{T^\alpha} \xrightarrow{p} 0.$$

As shown in the appendix, this also bounds the frequentist regret for any MDP with non-zero probability. State-of-the-art guarantees similar to Theorem 1 are satisfied by the algorithms UCRL2 [4] and REGAL [5] for the case of non-episodic RL. Here UCRL2 gives regret bounds $\tilde{O}(DS\sqrt{AT})$ where $D = \max_{s' \neq s} \min_\pi \mathbb{E}[T(s'|M, \pi, s)]$ and $T(s'|M, \pi, s)$ is the first time step where $s'$ is reached from $s$ under the policy $\pi$. REGAL improves this result to $\tilde{O}(\Psi S \sqrt{AT})$ where $\Psi \leq D$ is the span of the of the optimal value function. However, there is so far no computationally tractable implementation of this algorithm.

In many practical applications we may be interested in episodic learning tasks where the constants $D$ and $\Psi$ could be improved to take advantage of the episode length $\tau$. Simple modifications to both UCRL2 and REGAL will produce regret bounds of $\tilde{O}(\tau S \sqrt{AT})$, just as PSRL. This is close to the theoretical lower bounds of $\sqrt{SAT}$-dependence.

## 4 True versus sampled MDP

A simple observation, which is central to our analysis, is that, at the start of each $k$th episode, $M^*$ and $M_k$ are identically distributed. This fact allows us to relate quantities that depend on the true, but unknown, MDP $M^*$, to those of the sampled MDP $M_k$, which is

fully observed by the agent. We introduce $\sigma(H_{t_k})$ as the $\sigma$-algebra generated by the history up to $t_k$. Readers unfamiliar with measure theory can think of this as "all information known just before the start of period $t_k$." When we say that a random variable X is $\sigma(H_{t_k})$-measurable, this intuitively means that although X is random, it is deterministically known given the information contained in $H_{t_k}$. The following lemma is an immediate consequence of this observation [15].

**Lemma 1** (Posterior Sampling). *If $f$ is the distribution of $M^*$ then, for any $\sigma(H_{t_k})$-measurable function $g$,*

$$\mathbb{E}[g(M^*)|H_{t_k}] = \mathbb{E}[g(M_k)|H_{t_k}]. \tag{2}$$

Note that taking the expectation of (2) shows $\mathbb{E}[g(M^*)] = \mathbb{E}[g(M_k)]$ through the tower property.

Recall, we have defined $\Delta_k = \sum_{s \in \mathcal{S}} \rho(s)(V_{\mu^*,1}^{M^*}(s) - V_{\mu_k,1}^{M^*}(s))$ to be the regret over period $k$. A significant hurdle in analyzing this equation is its dependence on the optimal policy $\mu^*$, which we do not observe. For many reinforcement learning algorithms, there is no clean way to relate the unknown optimal policy to the states and actions the agent actually observes. The following result shows how we can avoid this issue using Lemma 1. First, define

$$\tilde{\Delta}_k = \sum_{s \in \mathcal{S}} \rho(s)(V_{\mu_k,1}^{M_k}(s) - V_{\mu_k,1}^{M^*}(s)) \tag{3}$$

as the difference in expected value of the policy $\mu_k$ under the sampled MDP $M_k$, which is known, and its performance under the true MDP $M^*$, which is observed by the agent.

**Theorem 2** (Regret equivalence).

$$\mathbb{E}\left[\sum_{k=1}^{m} \Delta_k\right] = \mathbb{E}\left[\sum_{k=1}^{m} \tilde{\Delta}_k\right] \tag{4}$$

*and for any $\delta > 0$ with probability at least $1 - \delta$,*

*Proof.* Note, $\Delta_k - \tilde{\Delta}_k = \sum_{s \in \mathcal{S}} \rho(s)(V_{\mu^*,1}^{M^*}(s) - V_{\mu_k,1}^{M_k}(s)) \in [-\tau, \tau]$. By Lemma 1, $\mathbb{E}[\Delta_k - \tilde{\Delta}_k | H_{t_k}] = 0$. Taking expectations of these sums therefore establishes the claim. □

This result bounds the agent's regret in epsiode $k$ by the difference between the agent's estimate $V_{\mu_k,1}^{M_k}(s_{t_k})$ of the expected reward in $M_k$ from the policy it chooses, and the expected reward $V_{\mu_k,1}^{M^*}(s_{t_k})$ in $M^*$. If the agent has a poor estimate of the MDP $M^*$, we expect it to learn as the performance of following $\mu_k$ under $M^*$ differs from its expectation under $M_k$. As more information is gathered, its performance should improve. In the next section, we formalize these ideas and give a precise bound on the regret of posterior sampling.

## 5 Analysis

An essential tool in our analysis will be the dynamic programming, or Bellman operator $\mathcal{T}_\mu^M$, which for any MDP $M = (\mathcal{S}, \mathcal{A}, R^M, P^M, \tau, \rho)$, stationary policy $\mu : \mathcal{S} \to \mathcal{A}$ and value function $V : \mathcal{S} \to \mathbb{R}$, is defined by

$$\mathcal{T}_\mu^M V(s) := \overline{R}_\mu^M(s, \mu) + \sum_{s' \in \mathcal{S}} P_{\mu(s)}^M(s'|s)V(s').$$

This operation returns the expected value of state $s$ where we follow the policy $\mu$ under the laws of $M$, for one time step. The following lemma gives a concise form for the dynamic programming paradigm in terms of the Bellman operator.

**Lemma 2** (Dynamic programming equation). *For any MDP $M = (\mathcal{S}, \mathcal{A}, R^M, P^M, \tau, \rho)$ and policy $\mu : \mathcal{S} \times \{1, \ldots, \tau\} \to \mathcal{A}$, the value functions $V_\mu^M$ satisfy*

$$V_{\mu,i}^M = \mathcal{T}_{\mu(\cdot,i)}^M V_{\mu,i+1}^M \tag{5}$$

*for $i = 1 \ldots \tau$, with $V_{\mu,\tau+1}^M := 0$.*

In order to streamline our notation we will let $V_{\mu,i}^* := V_{\mu,i}^{M^*}$, $V_{\mu,i}^k(s) := V_{\mu,i}^{M_k}(s)$, $\mathcal{T}_\mu^k := \mathcal{T}_\mu^{M_k}$, $\mathcal{T}_\mu^* := \mathcal{T}_\mu^{M^*}$ and $P_\mu^*(\cdot|s) := P_{\mu(s)}^{M^*}(\cdot|s)$.

## 5.1 Rewriting regret in terms of Bellman error

$$\mathbb{E}\left[\tilde{\Delta}_k\big|M^*, M_k\right] = \mathbb{E}\left[\sum_{i=1}^{\tau}\left[(\mathcal{T}_{\mu_k(\cdot,i)}^k - \mathcal{T}_{\mu_k(\cdot,i)}^*)V_{\mu_k,i+1}^k(s_{t_k+i})\right]\bigg|M^*, M_k\right] \qquad (6)$$

To see why (6) holds, simply apply the Dynamic programming equation inductively:

$$
\begin{aligned}
(V_{\mu_k,1}^k - V_{\mu_k,1}^*)(s_{t_k+1}) &= (\mathcal{T}_{\mu_k(\cdot,1)}^k V_{\mu_k,2}^k - \mathcal{T}_{\mu_k(\cdot,1)}^* V_{\mu_k,2}^*)(s_{t_k+1}) \\
&= (\mathcal{T}_{\mu_k(\cdot,1)}^k - \mathcal{T}_{\mu_k(\cdot,1)}^*)V_{\mu_k,2}^k(s_{t_k+1}) \\
&\quad + \sum_{s'\in\mathcal{S}}\{P_{\mu_k(\cdot,1)}^*(s'|s_{t_k+1})(V_{\mu_k,2}^* - V_{\mu_k,2}^k)(s')\} \\
&= (\mathcal{T}_{\mu_k(\cdot,1)}^k - \mathcal{T}_{\mu_k(\cdot,1)}^*)V_{\mu_k,2}^k(s_{t_k+1}) + (V_{\mu_k,2}^* - V_{\mu_k,2}^k)(s_{t_k+1}) + d_{t_k+1} \\
&= \dots \\
&= \sum_{i=1}^{\tau}(\mathcal{T}_{\mu_k(\cdot,i)}^k - \mathcal{T}_{\mu_k(\cdot,i)}^*)V_{\mu_k,i+1}^k(s_{t_k+i}) + \sum_{i=1}^{\tau}d_{t_k+i},
\end{aligned}
$$

where $d_{t_k+i} := \sum_{s'\in\mathcal{S}}\{P_{\mu_k(\cdot,i)}^*(s'|s_{t_k+i})(V_{\mu_k,i+1}^* - V_{\mu_k,i+1}^k)(s')\} - (V_{\mu_k,i+1}^* - V_{\mu_k,i+1}^k)(s_{t_k+i})$.

This expresses the regret in terms two factors. The first factor is the one step *Bellman error* $\left[(\mathcal{T}_{\mu_k(\cdot,i)}^k - \mathcal{T}_{\mu_k(\cdot,i)}^*)V_{\mu_k,i+1}^k(s_{t_k+i})\right]$ under the sampled MDP $M_k$. Crucially, (6) depends only the Bellman error under the observed policy $\mu_k$ and the states $s_1, .., s_T$ that are actually visited over the first $T$ periods. We go on to show the posterior distribution of $M_k$ concentrates around $M^*$ as these actions are sampled, and so this term tends to zero.

The second term captures the randomness in the transitions of the true MDP $M^*$. In state $s_t$ under policy $\mu_k$, the expected value of $(V_{\mu_k,i+1}^* - V_{\mu_k,i+1}^k)(s_{t_k+i})$ is exactly $\sum_{s'\in\mathcal{S}}\{P_{\mu_k(\cdot,i)}^*(s'|s_{t_k+i})(V_{\mu_k,i+1}^* - V_{\mu_k,i+1}^k)(s')\}$. Hence, conditioned on the true MDP $M^*$ and the sampled MDP $M_k$, the term $\sum_{i=1}^{\tau}d_{t_k+i}$ has expectation zero.

## 5.2 Introducing confidence sets

The last section reduced the algorithm's regret to its expected Bellman error. We will proceed by arguing that the sampled Bellman operator $\mathcal{T}_{\mu_k(\cdot,i)}^k$ concentrates around the true Bellman operatior $\mathcal{T}_{\mu_k(\cdot,i)}^*$. To do this, we introduce high probability confidence sets similar to those used in [4] and [5]. Let $\hat{P}_a^t(\cdot|s)$ denote the empirical distribution up period $t$ of transitions observed after sampling $(s,a)$, and let $\hat{R}_a^t(s)$ denote the empirical average reward. Finally, define $N_{t_k}(s,a) = \sum_{t=1}^{t_k-1}\mathbb{1}_{\{(s_t,a_t)=(s,a)\}}$ to be the number of times $(s,a)$ was sampled prior to time $t_k$. Define the confidence set for episode $k$:

$$\mathcal{M}_k := \left\{M : \left\|\hat{P}_a^t(\cdot|s) - P_a^M(\cdot|s)\right\|_1 \le \beta_k(s,a) \ \& \ |\hat{R}_a^t(s) - R_a^M(s)| \le \beta_k(s,a) \ \forall(s,a)\right\}$$

Where $\beta_k(s,a) := \sqrt{\frac{14S\log(2SAmt_k)}{\max\{1,N_{t_k}(s,a)\}}}$ is chosen conservatively so that $\mathcal{M}_k$ contains both $M^*$ and $M_k$ with high probability. It's worth pointing out that we have not tried to optimize this confidence bound, and it can be improved, at least by a numerical factor, with more careful analysis. Now, using that $\tilde{\Delta}_k \le \tau$ we can decompose regret as follows:

$$\sum_{k=1}^{m}\tilde{\Delta}_k \le \sum_{k=1}^{m}\tilde{\Delta}_k\mathbb{1}_{\{M_k,M^*\in\mathcal{M}_k\}} + \tau\sum_{k=1}^{m}[\mathbb{1}_{\{M_k\notin\mathcal{M}_k\}} + \mathbb{1}_{\{M^*\notin\mathcal{M}_k\}}] \qquad (7)$$

Now, since $\mathcal{M}_k$ is $\sigma(H_{t_k})$-measureable, by Lemma 1, $\mathbb{E}[\mathbb{1}_{\{M_k \notin \mathcal{M}_k\}}|H_{t_k}] = \mathbb{E}[\mathbb{1}_{\{M^* \notin \mathcal{M}_k\}}|H_{t_k}]$. Lemma 17 of [4] shows[3] $\mathbb{P}(M^* \notin \mathcal{M}_k) \leq 1/m$ for this choice of $\beta_k(s,a)$, which implies

$$
\begin{aligned}
\mathbb{E}\left[\sum_{k=1}^{m} \tilde{\Delta}_k\right] &\leq \mathbb{E}\left[\sum_{k=1}^{m} \tilde{\Delta}_k \mathbb{1}_{\{M_k, M^* \in \mathcal{M}_k\}}\right] + 2\tau \sum_{k=1}^{m} \mathbb{P}\{M^* \notin \mathcal{M}_k\}. \\
&\leq \mathbb{E}\left[\sum_{k=1}^{m} \mathbb{E}\left[\tilde{\Delta}_k | M^*, M_k\right] \mathbb{1}_{\{M_k, M^* \in \mathcal{M}_k\}}\right] + 2\tau \\
&\leq \mathbb{E}\sum_{k=1}^{m}\sum_{i=1}^{\tau} |(\mathcal{T}_{\mu_k(\cdot,i)}^k - \mathcal{T}_{\mu_k(\cdot,i)}^*)V_{\mu_k,i+1}^k(s_{t_k+i})| \mathbb{1}_{\{M_k, M^* \in \mathcal{M}_k\}} + 2\tau \\
&\leq \tau \mathbb{E}\sum_{k=1}^{m}\sum_{i=1}^{\tau} \min\{\beta_k(s_{t_k+i}, a_{t_k+i}), 1\} + 2\tau. \qquad (8)
\end{aligned}
$$

We also have the worst–case bound $\sum_{k=1}^{m} \tilde{\Delta}_k \leq T$. In the technical appendix we go on to provide a worst case bound on $\min\{\tau \sum_{k=1}^{m}\sum_{i=1}^{\tau} \min\{\beta_k(s_{t_k+i}, a_{t_k+i}), 1\}, T\}$ of order $\tau S \sqrt{AT \log(SAT)}$, which completes our analysis.

## 6  Simulation results

We compare performance of PSRL to UCRL2 [4]: an optimistic algorithm with similar regret bounds. We use the standard example of *RiverSwim* [21], as well as several randomly generated MDPs. We provide results in both the episodic case, where the state is reset every $\tau = 20$ steps, as well as the setting without episodic reset.

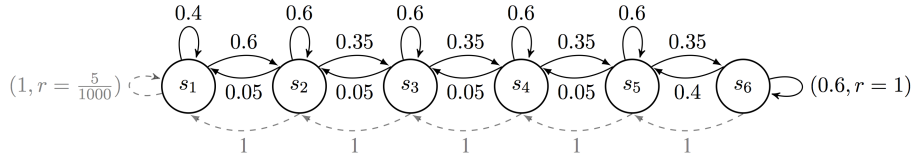

Figure 1: *RiverSwim* - continuous and dotted arrows represent the MDP under the actions "right" and "left".

*RiverSwim* consists of six states arranged in a chain as shown in Figure 1. The agent begins at the far left state and at every time step has the choice to swim left or right. Swimming left (with the current) is always successful, but swimming right (against the current) often fails. The agent receives a small reward for reaching the leftmost state, but the optimal policy is to attempt to swim right and receive a much larger reward. This MDP is constructed so that efficient exploration is required in order to obtain the optimal policy. To generate the random MDPs, we sampled 10-state, 5-action environments according to the prior.

We express our prior in terms of Dirichlet and normal-gamma distributions over the transitions and rewards respectively.[4] In both environments we perform 20 Monte Carlo simulations and compute the total regret over 10,000 time steps. We implement UCRL2 with $\delta = 0.05$ and optimize the algorithm to take account of finite episodes where appropriate. PSRL outperformed UCRL2 across every environment, as shown in Table 1. In Figure 2, we show regret through time across 50 Monte Carlo simulations to 100,000 time–steps in the *RiverSwim* environment: PSRL's outperformance is quite extreme.

Table 1: Total regret in simulation. PSRL outperforms UCRL2 over different environments.

| Algorithm | Random MDP $\tau$-episodes | Random MDP $\infty$-horizon | RiverSwim $\tau$-episodes | RiverSwim $\infty$-horizon |
|---|---|---|---|---|
| PSRL | $1.04 \times 10^4$ | $7.30 \times 10^3$ | $6.88 \times 10^1$ | $1.06 \times 10^2$ |
| UCRL2 | $5.92 \times 10^4$ | $1.13 \times 10^5$ | $1.26 \times 10^3$ | $3.64 \times 10^3$ |

## 6.1 Learning in MDPs without episodic resets

The majority of practical problems in reinforcement learning can be mapped to repeated episodic interactions for some length $\tau$. Even in cases where there is no actual reset of episodes, one can show that PSRL's regret is bounded against all policies which work over horizon $\tau$ or less [6]. Any setting with discount factor $\alpha$ can be learned for $\tau \propto (1-\alpha)^{-1}$.

One appealing feature of UCRL2 [4] and REGAL [5] is that they learn this optimal timeframe $\tau$. Instead of computing a new policy after a fixed number of periods, they begin a new episode when the total visits to any state-action pair is doubled. We can apply this same rule for episodes to PSRL in the $\infty$-horizon case, as shown in Figure 2. Using optimism with KL-divergence instead of $L^1$ balls has also shown improved performance over UCRL2 [22], but its regret remains orders of magnitude more than PSRL on *RiverSwim*.

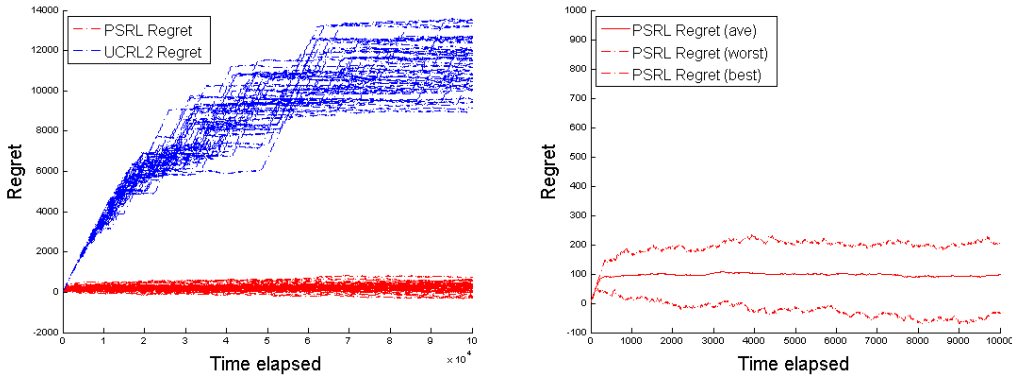

(a) PSRL outperforms UCRL2 by large margins (b) PSRL learns quickly despite misspecified prior

Figure 2: Simulated regret on the $\infty$-horizon *RiverSwim* environment.

## 7 Conclusion

We establish *posterior sampling for reinforcement learning* not just as a heuristic, but as a provably efficient learning algorithm. We present $\tilde{O}(\tau S\sqrt{AT})$ Bayesian regret bounds, which are some of the first for an algorithm not motivated by optimism and are close to state of the art for any reinforcement learning algorithm. These bounds hold in expectation irrespective of prior or model structure. PSRL is conceptually simple, computationally efficient and can easily incorporate prior knowledge. Compared to feasible optimistic algorithms we believe that PSRL is often more efficient statistically, simpler to implement and computationally cheaper. We demonstrate that PSRL performs well in simulation over several domains. We believe there is a strong case for the wider adoption of algorithms based upon posterior sampling in both theory and practice.

**Acknowledgments**

Osband and Russo are supported by Stanford Graduate Fellowships courtesy of PACCAR inc., and Burt and Deedee McMurty, respectively. This work was supported in part by Award CMMI-0968707 from the National Science Foundation.

## Footnotes

[1]For an MDP, this might be a prior over transition dynamics and reward distributions.

[2]We alter terminology since PSRL is neither Bayes-optimal, nor a direct approximation of this.

[3]Our confidence sets are equivalent to those of [4] when the parameter $\delta = 1/m$.

[4]These priors are conjugate to the multinomial and normal distribution. We used the values $\alpha = 1/S, \mu = \sigma^2 = 1$ and pseudocount $n = 1$ for a diffuse uniform prior.

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

# A   Relating Bayesian to frequentist regret

Let $\mathcal{M}$ be any family of MDPs with non-zero probability under the prior. Then, for any $\epsilon > 0$, $\alpha > \frac{1}{2}$:

$$\mathbb{P}\left(\frac{\text{Regret}(T, \pi_\tau^{PS})}{T^\alpha} > \epsilon \Big| M^* \in \mathcal{M}\right) \to 0$$

This provides regret bounds even if $M^*$ is not distributed according to $f$. As long as the true MDP is not impossible under the prior, we will have an asymptotic frequentist regret close to the theoretical lower bounds of in $T$-dependence of $O(\sqrt{T})$.

*Proof.* We have for any $\epsilon > 0$:

$$\frac{\mathbb{E}[\text{Regret}(T, \pi_\tau^{PS})]}{T^\alpha} \quad \geq \quad \mathbb{E}\left[\frac{\text{Regret}(T, \pi_\tau^{PS})}{T^\alpha}\Big| M^* \in \mathcal{M}\right]\mathbb{P}\left(M^* \in \mathcal{M}\right)$$

$$\geq \quad \epsilon \mathbb{P}\left(\frac{\text{Regret}(T, \pi_\tau^{PS})}{T^\alpha}\Big| M^* \in \mathcal{M}\right)\mathbb{P}\left(M^* \in \mathcal{M}\right)$$

Therefore via theorem (1), for any $\alpha > \frac{1}{2}$:

$$\mathbb{P}\left(\frac{\text{Regret}(T, \pi_\tau^{PS})}{T^\alpha}\Big| M^* \in \mathcal{M}\right) \leq \left(\frac{1}{\epsilon \mathbb{P}\left(M^* \in \mathcal{M}\right)}\right)\frac{\mathbb{E}[\text{Regret}(T, \pi^P S_\tau)]}{T^\alpha} \to 0$$

$\square$

# B   Bounding the sum of confidence set widths

We are interested in bounding $\min\{\tau \sum_{k=1}^m \sum_{i=1}^\tau \min\{\beta_k s_{t_k+i}, a_{t_k+i}\}, 1\}, T\}$ which we claim is $O(\tau S\sqrt{AT\log(SAT)})$ for $\beta_k(s,a) := \sqrt{\frac{14S\log(2SAmt_k)}{\max\{1, N_{t_k}(s,a)\}}}$.

*Proof.* In a manner similar to [4] we can say:

$$\sum_{k=1}^m \sum_{i=1}^\tau \sqrt{\frac{14S\log(2SAmt_k)}{\max\{1, N_{t_k}(s,a)\}}} \quad \leq \quad \sum_{k=1}^m \sum_{i=1}^\tau \mathbb{1}_{\{N_{t_k} \leq \tau\}} + \sum_{k=1}^m \sum_{i=1}^\tau \mathbb{1}_{\{N_{t_k} > \tau\}}\sqrt{\frac{14S\log(2SAmt_k)}{\max\{1, N_{t_k}(s,a)\}}}$$

Now, the consider the event $(s_t, a_t) = (s,a)$ and $(N_{t_k}(s,a) \leq \tau)$. This can happen fewer than $2\tau$ times per state action pair. Therefore, $\sum_{k=1}^m \sum_{i=1}^\tau \mathbf{1}(N_{t_k}(s,a) \leq \tau) \leq 2\tau SA$. Now, suppose $N_{t_k}(s,a) > \tau$. Then for any $t \in \{t_k, .., t_{k+1} - 1\}$, $N_t(s,a) + 1 \leq N_{t_k}(s,a) + \tau \leq 2N_{t_k}(s,a)$. Therefore:

$$\sum_{k=1}^m \sum_{t=t_k}^{t_{k+1}-1} \sqrt{\frac{\mathbf{1}(N_{t_k}(s_t, a_t) > \tau)}{N_{t_k}(s_t, a_t)}} \quad \leq \quad \sum_{k=1}^m \sum_{t=t_k}^{t_{k+1}-1} \sqrt{\frac{2}{N_t(s_t, a_t) + 1}} = \sqrt{2}\sum_{t=1}^T (N_t(s_t, a_t) + 1)^{-1/2}$$

$$\leq \quad \sqrt{2}\sum_{s,a}\sum_{j=1}^{N_{T+1}(s,a)} j^{-1/2} \leq \sqrt{2}\sum_{s,a}\int_{x=0}^{N_{T+1}(s,a)} x^{-1/2}\, dx$$

$$\leq \quad \sqrt{2SA\sum_{s,a} N_{T+1}(s,a)} = \sqrt{2SAT}$$

Note that since all rewards and transitions are absolutely constrained $\in [0, 1]$ our regret

$$\min\{\tau \sum_{k=1}^m \sum_{i=1}^\tau \min\{\beta_k(s_{t_k+i}, a_{t_k+i}), 1\}, T\} \quad \leq \quad \min\{2\tau^2 SA + \tau\sqrt{28S^2AT\log(SAT)}, T\}$$

$$\leq \quad \sqrt{2\tau^2 SAT} + \tau\sqrt{28S^2AT\log(SAT)} \leq \tau S\sqrt{30AT\log(SAT)}$$

Which is our required result. $\square$

