[Reviews · NeurIPS 2013]

Submitted by Assigned_Reviewer_7

Title: (More) Efficient Reinforcement Learning via Posterior Sampling

Summary: The paper proposes a Thompson sampling-type approach for episodic finite-horizon reinforcement learning, called posterior sampling for reinforcement learning. A regret analysis showing state of the art performance is provided, as well as simulations showing better empirical performance on some toy problems than other algorithms with similar regret bounds.

Comments:

The paper is clear and very well written. The contribution is also clearly identified: showing that is possible to use a Thompson sampling-type approach in a RL setting, and also theoretically analyse that is can be efficient.

Regarding the regret bound, the authors say that the regret bound of PSRL is close to state of the art. It would be nice to recall in this paper what are such state-of-the art values.

This is a rather frequent question in the field of multi-armed bandit algorithms, but how large is the constant in practice (in the O())?;

Comparisons could be made with the algorithm KL-UCRL (Optimism in reinforcement learning and Kullback-Leibler divergence - S. Filippi, O. Cappé, A. Garivier), an algorithm based on the optimism principle, which was proven to have better empirical performance than UCRL2 on the riverSwim environment, with a regret in $\tilde O(|S|\sqrt{|A&T})$ as well;

Minor comments:
- l.99: it seems that the $s_1$ should be replaced by a $s_i$;
- l.192: "we how we can" -> how we can;
- l.434: "markov" -> Markov;
- references should be uniformed (Firstname Name, F. Name, etc)
Summary: This is a technically strong, well-written and interesting paper.

Submitted by Assigned_Reviewer_8

Summary: this paper gives a regret analysis for an episodic reinforcement learning motivated by posterior sampling (as known as Thompson sampling). The idea is to sample, from the posterior, an MDP model at the beginning of each episode, follow this MDP's optimal policy in that episode, update the model posterior, and then repeat. In the past, despite good empirical performance, this algorithm (called PSRL) remains a heuristic approach without strong theoretical guarantees. This paper provides the first regret analysis, with a bound of O(\tau * S * \sqrt{A * T}), where H is the length of the episode and T is the total number of stpes. Numerical experiments on very simple MDPs show PSRL significantly outperform the other algorithm UCRL2 that enjoys a similar regret bound.

* Quality: this paper makes an interesting theoretical contribution to RL in finite-state, finite-action, episodic MDPs. Building on recent advances in the multi-armed bandit literature, it is a successful step towards understanding finite-sample performance of Bayesian-style RL algorithms.

* Clarity: the paper is written very clearly. Despite the heavy technical details, the basic ideas are explained well and intuitively.

* Originality: The algorithm PSRL is not new; instead, the novelty in the paper is in its (first) regret analysis. The key in the analysis is Lemma 2, which equates the *expected* regret with the regret in the sampled MDP (assuming the MDP is sampled from the posterior). A similar observation has been used to proved expected regret bounds of posterior sampling in multi-armed bandits [15]. While the rest of the analysis seems pretty standard in reinforcement learning, the application of Lemma 2 in this kind of analysis, and especially for expected regret, appears sufficiently novel.

* Significance: Posterior sampling has been very successful in multi-armed bandits that raised quite a lot of interests in its theoretical study recently. This paper is the first that extends analytic ideas to MDPs and shows strong regret bounds of posterior sampling. Not only is the result the first of its kind, it may generate further interests in the RL community to investigate a new class of algorithms based on posterior sampling.

Detailed comments:

* While the analysis and bounds are very interesting, the paper should make it clear that the nature of the bounds here is fundamentally different from that in the literature. In particular, existing bounds are worst-case bounds, while the bounds here are average-case (averaged by the prior/posterior). It is the averaging that makes the key Lemma 2 possible. For the same reason, it seems the paper's *expected* bounds may be weaker than worst-case bounds. I hope the authors can comment on these points in the paper.

* In the paper, UCRL2 is the only algorithm to compare against PSRL. Justifications are needed for this choice. PSRL, in the current description in the paper, applies to episodic tasks and aims to optimize undiscounted, finite-horizon total reward. In contrast, UCRL2 applies to continuing tasks and aims to optimize average reward. As a consequence, their regret bounds are not directly comparable: UCRL2 depends on a diameter parameter, while PSRL has the horizon length in the bound. There may be a way to relate them, but it is not obvious how.

* Instead of UCRL2, another piece of work may be more relevant:
Claude-Nicolas Fiechter: Efficient Reinforcement Learning. COLT 1994: 88-97
Although the Fiechter paper considers the PAC framework, it considers finite-horizon problems, so seems relevant to the present work.

* The simple experiments show PSRL is empirically much better than UCRL2. It does not strike me as a surprise, given the very conservative nature of UCRL2. And, because of the gap the kinds of problems the two algorithms are designed for (see a related comment above), the empirical comparison carries even less information.

* Line 255, 'depends only the' --> 'depends only on the'

* Line 278, please be specific what the 'high probability' is.

* Line 411, is it too restrictive to assume that the optimal average reward is uncorrelated with episodic length?
Summary: An interesting analysis for an important algorithm, although the regret bounds are not directly comparable to previous results. The paper is almost purely theoretical, with limited experiments in simple MDPs.

Submitted by Assigned_Reviewer_10

This paper provides the first regret bounds for a reinforcement learning algorithm that 1) maintains a posterior over the MDP, 2) samples one MDP from the prior, and 3) follows the policy that is optimal for that one MDP. This kind of "Posterior-Sampling Reinforcement Learning" (PSRL) algorithm has been proposed before, but without regret bounds. All algorithms with regret bounds are based on the principle of optimism in the face of uncertainty. This is the first regret bound analysis of an algorithm not based on this principle. This algorithm's bound is not quite as tight as those based on this principle, but in practice in typical cases it is substantially more exploitive, and appears to be much more efficient overall.

If it is correct that this is the first regret result for such an algorithm (not based on optimism), and if this result is correct, then the paper should be accepted.

The paper is written with admirable clarity. Nevertheless, I did not understand the proof in detail, so I may have missed errors.

For simplicity, the work assumes an episodic formulation with fixed-length episodes. This is unrealistic, but is ok for a result that breaks new ground such as this one. This and other aspects of the algorithm that are undesirable in practice can (probably) be removed in future work if the main result shown here holds up. One related idea is discussed in a later section of the paper. This is good, but it could be omitted if it made room to include what is asked for in the paragraph below.

The paper claims that the regret bound for its algorithm is close to the state of the art, but does not state what the state of the art is. It should do this, both as part of making its claim clear and allowing the reader to judge that the new algorithm's bound is in fact close. The paper should discuss the sense in which the new algorithm's bound is close.

Please don't include citations as parts of sentences.
Summary: The first regret bound, and promising empirical results, for a reinforcement learning algorithm based on posterior sampling rather than optimism in the face of uncertainty.
Author Feedback

Author rebuttal: Thank you for your feedback, it has been very helpful. We will try to address the main points as concisely as possible:

*** What are the state-of-the-art regret bounds ***
UCLR2 gives O(DS\sqrt{AT}), REGAL gives a tighter bound with D replaced by span (the span-seminorm of the optimal value function), but REGAL is not computationally tractable. For the episodic case, a simple alteration to UCRL2 to change policy only every episode would give an O(tauS\sqrt{AT}) bound, just as per PSRL. We believe this bound to be the state-of-the-art and will include it in the revision.

*** PSRL is for finite episodes, UCRL2 is infinite interaction ***
We are not yet able to extend our analysis of PSRL to the infinite undiscounted interaction of UCRL2. However, we believe that the finite episode case, and the closely-related discounted problem, is of practical interest to many applications. For these applications we have compared PSRL to a UCRL2 variant designed for finite episodes, where policies are recalculated only at the start of every episode and designed to maximize episodic performance. We also compare full UCRL2 to a PSRL variant using the exponentially growing episodes of UCRL2, which performs very well. We mention in line 411 that under some (possibly restrictive) assumptions we can present analogous results for the infinite horizon case.

*** Are the regret bounds given for PSRL comparable to those of UCRL2 ***
There is some distinction that Theorem 1 gives a bound on regret when taken over a Bayesian prior, generally this is not as strong as those of UCRL2. Our bounds are of expected regret (BayesRisk) as opposed to worst-case MDP bounds with high probability. Corollary 1 shows that these two notions are closely related in T-dependence. It should be noted that these bounds are given with respect to a prior, so that if the prior is “bad” then so might the bounds. We have attempted to clarify this point.

*** Why do we only compare performance to UCRL2 ***
UCRL2 holds similar expected regret bounds to PSRL and their analysis is very similar. We had initially intended to include the KL-UCRL variant, which has been shown to perform marginally better than UCRL in practice. However, due to space constraints, difficulties in replicating empirical results of prior work on KL-UCRL, and the fact that PSRL is orders of magnitude more effective than previous claims of KL-UCRL, we decided to keep things simple.

*** Is Fiechter: Efficient Reinforcement Learning more relevant ***
This paper also considers the finite-horizon case of reinforcement learning, with analysis is presented in the PAC sense, as opposed to regret bounds. Despite the episodic setting, we feel the algorithm's approach, separating exploitation from exploration explicitly and with fundamentally different guarantees, is more similar to other PAC algorithms like R_max than PSRL. It might be interesting to compare the two in terms of performance, or address the PAC vs regret distinction, although we are pressed for space. For reasons outlined earlier in this response we believe that UCRL2 offers a more natural comparison for PSRL.

*** What is the constant in the O() regret ***
Our analysis gives a leading order coefficient of 4\sqrt{14} which is similar to the 2\sqrt{14}(\sqrt{2}+1) given in UCRL2. The lower order terms are sometimes 2x as big for PSRL vs UCRL2, since we have to account for both the sampled and actual MDP in the analysis. The general picture is that it is very similar to UCRL2 analysis. We are able to refine these bounds at least to \sqrt{10} by decreasing the size of the \beta confidence sets. We decided not to do this for ease/brevity of maintaining UCRL2’s analysis which would be familiar to many.